# Genome-Wide Identification and Characterization of Circular RNAs during Skeletal Muscle Development in Meat Rabbits

**DOI:** 10.3390/ani12172208

**Published:** 2022-08-27

**Authors:** Kun Du, Xiaoyu Zhao, Yanhong Li, Zhoulin Wu, Wenqiang Sun, Jie Wang, Xianbo Jia, Shiyi Chen, Songjia Lai

**Affiliations:** Farm Animal Genetic Resources Exploration and Innovation Key Laboratory of Sichuan Province, Sichuan Agricultural University, Chengdu 611130, China

**Keywords:** whole-transcriptome sequencing, circRNA-seq, ZIKA, networks, ceRNAs

## Abstract

**Simple Summary:**

Our knowledge of circRNAs regulating skeletal muscle development remains largely unknown in meat rabbits. Therefore, we collected the leg muscle tissues of ZIKA rabbits at three key growth stages. A combination of circRNA assembly from a circRNA-seq library and the whole-transcriptome sequencing data identified credible circRNAs in our samples. We found these circRNAs were more conserved between rabbits and humans than between rabbits and mice. A prediction of circRNA–microRNA–mRNAs networks revealed that circRNAs might be the regulators that mainly functioned in rabbits’ muscle neuron development and metabolic processes. Our work provides a catalog of circRNAs regulating skeletal muscle development at key growth stages in rabbits and might give a new insight into rabbit breeding.

**Abstract:**

Skeletal muscle development plays a vital role in muscle quality and yield in meat rabbits. Circular RNAs (circRNAs) are a new type of single-stranded endogenous non-coding RNAs involved in different biological processes. However, our knowledge of circRNAs regulating skeletal muscle development remains largely unknown in meat rabbits. In this study, we collected the leg muscle tissues of ZIKA rabbits at three key growth stages. By performing whole-transcriptome sequencing, we found the sequential expression of day 0- (D0-), D35-, and D70-selective mRNAs mainly functioned in muscle development, nervous development, and immune response during skeletal muscle development, respectively. Then, a combination of circRNA assembly from a circRNA-seq library and the whole-transcriptome sequencing data identified 6845 credible circRNAs in our samples. Most circRNAs were transcribed from exons of known genes, contained few exons, and showed short length, and these circRNAs were more conserved between rabbits and humans than between rabbits and mice. The upregulated circRNAs, which were synchronously changed with host genes, primarily played roles in MAPK signaling pathways and fatty acid biosynthesis. The prediction of circRNA–microRNA–mRNAs networks revealed that circRNAs might be the regulators that mainly functioned in rabbits’ muscle neuron development and metabolic processes. Our work provides a catalog of circRNAs regulating skeletal muscle development at key growth stages in rabbits and might give a new insight into rabbit breeding.

## 1. Introduction

Due to the rich protein, low fat, and low cholesterol levels, rabbit (Oryctolagus cuniculus) meat has become increasingly popular among people [1]. Skeletal muscle development plays a vital role in muscle quality and yield, which is a key factor affecting the economic value of rabbits and a major target trait in the breeding program of meat rabbits [2]. The postnatal growth of skeletal muscle is mainly achieved by increasing the length and circumference of muscle fibers in rabbits. Biotechnologies and functional genomic theories currently enable breeders to decipher molecular mechanisms of skeletal muscle development. DNA methylation, genomic structural variation, and microRNAs (miRNAs) have been reported to regulate muscle development at different genetic levels [3,4,5]. Nevertheless, our understanding of the genetic regulation of skeletal muscle development remains at its infancy.

Circular RNAs (circRNAs) are a new type of single-stranded endogenous non-coding RNAs, which lack 5′ and 3′ structures, and form covalently closed loops, and are highly conserved across species [6,7]. Previous studies have indicated that circRNAs can act as competing endogenous RNAs to mRNAs via the shared interaction miRNA between circRNAs and mRNAs, which regulate different biological processes such as cell differentiation, immune response, and cancer formation [8,9,10]. On the other hand, emerging data indicated that circRNAs could directly or indirectly interact with host genes to regulate host gene expression [11,12]. In domestic animals, circRNAs were reported to be involved in muscle development in pigs [13,14], goats [15], and bovines [16]. However, our knowledge of circRNAs regulating skeletal muscle development remains largely unknown in meat rabbits.

Legs are the main depots of skeletal muscle growth in rabbits [17,18]. The ZIKA rabbits are one of the famous meat rabbit lines in the world, with excellent meat production performance [19]. The present study aims to discover and characterize the genome-wide circRNAs of ZIKA rabbit skeletal muscle tissues and then reveal potential circRNA functions and involved signal pathways.

## 2. Materials and Methods

### 2.1. Ethics Approval

All surgical procedures involving rabbits were performed according to the approved protocols of the Biological Studies Animal Care and Use Committee, Sichuan Province, China. Rabbits had free access to food and water under normal conditions and were humanely sacrificed as necessary to ameliorate suffering (Ethics Code: DKY2020102011).

### 2.2. Sample Preparation

In this study, the ZIKA rabbits were raised at the breeding center of Sichuan Agricultural University, Ya’an, China. These rabbits were fed a standard diet as described in our previous study [20] and water ad libitum. Nine female rabbits at day 0 (D0, just after birth), D35, and D70 (three individuals per stage) were selected to dissect, which represented the newborn, weaning, and puberty stage, respectively. The rabbits were euthanized by injecting air into the ear vein after fully anesthetizing. Skeletal muscle tissues were isolated from the posterior of the right hind thigh of the rabbits. The samples were snap frozen in liquid nitrogen and then stored at −80 °C until RNA extraction.

### 2.3. Whole-Transcriptome Sequencing and Differential Analysis of mRNAs

In this study, the skeletal muscle tissues were subjected to whole-transcriptome sequencing. Total RNA was extracted using Trizol Reagent (Invitrogen, Hong Kong, China). The RNA concentration and purity were checked using Nanodrop2000 (ThermoFisher, Carlsbad, CA, USA), and the RNA integrity was checked using Agilent 2100 biological analyzer (Agilent Technologies, Carlsbad, CA, USA). The RNA samples with concentration >100 ng/μL, optical density of 260:280 ratio > 1.9, and RNA integrity number (RIN) value >8.0 were collected to construct libraries. Approximately 2 μg total RNA was subjected to construct whole-transcriptome sequencing libraries using the NR603-VAHTS Total RNA-Seq Library Prep Kit (Vazyme Biotech Co., Ltd., Nanjing, China). Briefly, RNA samples were fragmented to 200–500 bp after removing rRNA. Then, the fragmented RNA was transcribed to first-strand cDNA. The second strand of cDNA was synthesized using the dUTP method. After adding sequencing adapters, the USER enzyme was used to degrade the cDNA strands that contained the U base. The cDNA was amplified using a polymerase chain reaction (PCR), and the purified PCR products were used for RNA sequencing on an Illumina HiSeq2000 platform. Finally, 150 bp pair-end reads were generated.

The whole-transcriptome sequencing data of the 12 samples were subjected to analyzing the mRNA expression. The low-quality reads (reads containing 10% N base or reads that had more than 50% quality < 10 bases) and adapter sequences of raw data were filtered using Fastp software [21]. The quality check of clean reads was performed using Fastqc [22]. The rabbit genome and genomic annotation were downloaded from the ENSEMBL website (OryCun2.0, ENSEMBL release 101, https://asia.ensembl.org/index.html, accessed on 4 November 2021). Clean reads were mapped back to the rabbit genome using Hisat2 [23]. The raw read counts of mRNA were subjected to a differential analysis using DEseq2 [24]. The expression levels of mRNA were normalized using the value of counts per million (CPM) in DEseq2. The mRNAs with the thresholds of ‘|log2(fold-change)| > 1’ and ‘*p*-adj < 0.05’ were considered differentially expressed mRNAs (DEmRNAs).

### 2.4. Identification of circRNAs from Whole-Transcriptome Sequencing Data

The clean reads of datasets that passed the quality check were mapped back to the rabbit genome (OryCun2.0) using bowtie2 software with parameters of ‘—very-sensitive —score-min = C, −15, 0’, and unmapped reads were selected to the downstream circRNA identification [25]. The circRNA identification was performed using Find_circ according to the manufacturer’s recommendation [26], and the identified circRNAs with at least two back-splicing junction reads (BSJ reads) in one sample were used.

### 2.5. Identification and Assembly of Spliced Sequences of circRNA Using circRNA-seq

The existence of both linear and circular transcripts in the whole-transcriptome sequencing library could increase the ratio of false positive circRNA when performing the transcript assembly of circRNAs [27]. To obtain well-assembled spliced circRNA sequences, we performed circRNA-seq for a pooled RNA sample. Briefly, RNA samples extracted from individual samples were pooled, and RNase R (Epicentre, Madison, WI, USA) was used to remove linear RNA from the pooled sample. Then, the RNA sample enriched in circRNAs was used for circRNA-seq library preparation. The circRNA-seq library was prepared using NEBNext^®^ Ultra™ Directional RNA Library Prep Kit for Illumina^®^ (NEB, Carlsbad, CA, USA) according to the manufacturer’s recommendations. After adding sequencing adapters, the cDNA segments with 250–300 bp length were purified using the AMPure XP system (Beckman Coulter, Beverly, MA, USA). The purified segments underwent PCR amplification, PCR product purification, and library quality check. Finally, the quantified library was sequenced on an Illumina platform, and 150 bp paired-end reads were generated.

The adapter sequences and low-quality reads were removed using Fastp software [21]. The clean reads were mapped back to the rabbit genome using the Hisat2 program with default parameters, and a sequence alignment/map (SAM) file of reads was generated [23]. Then, Find_circ [26] and CIRI2 [28] were used to identify and assemble circRNA sequences in the SAM file, and the circRNAs that were detected by both types of software were used. Human and mouse circRNAs were retrieved from circBase (http://www.circbase.org/, accessed on 4 November 2021) and used to construct local BLAST databases using the ‘makeblastdb’ program [29]. Then, the sequences of circRNAs identified in the rabbit muscles were mapped to the databases. The circRNAs between the two species had a sequence hit with thresholds of ‘evalue < 1 × 10^−10^’, ‘pident > 90%’, and ‘total alignment length > 75% of total circRNA length’; these were considered as conserved circRNAs.

### 2.6. Differential Analysis of circRNAs

The circRNAs simultaneously identified from whole-transcriptome sequencing and circRNA-seq libraries were considered credible circRNAs and used for the downstream analysis. The raw read counts of BJS, estimated from whole-transcriptome data, were normalized using the CPM method and used for a differential analysis using DEseq2 software [24]. The circRNAs with thresholds of ‘|log2(fold-change)|> 1’ and ‘*p*-value < 0.05’ were considered as differentially expressed circRNAs (DECs).

### 2.7. CircRNA-miRNA-mRNA Network Construction

The differentially expressed miRNAs during the skeletal muscle development of rabbits were retrieved from a previous study conducted by Jing and colleagues [30]. The sequences of mature miRNA were retrieved from miRbase [31]. Then, the spliced sequences of circRNAs and mature sequences of miRNAs were used to predict the binding sites between circRNAs and miRNAs using miRanda software [32]. The interaction between circRNAs and miRNAs with ‘tot score > 150’ and ‘energy < −15’ were selected to construct circRNA-miRNA interaction networks using Cytoscape [33]. The 3′ UTR sequences of DEmRNAs, which were downloaded from the ENSEMBL website (https://www.ensembl.org, accessed on 4 November 2021), were used to predict interactions between miRNAs and mRNAs using miRanda software with parameters of ‘tot score > 150’ and ‘energy < −15’.

### 2.8. Functional Annotation

The gene ontology (GO) enrichment and KEGG pathway analyses were performed using R package Clusterprofilers [34]. The enriched GO terms and KEGG pathways with *p*-value < 0.05 were considered significant.

### 2.9. Statistical Analysis

Statistical analyses, including a *t*-test and one-way ANOVA, were conducted on R software. The *p*-value < 0.05 was considered significant.

## 3. Results

### 3.1. Differential Analysis of mRNAs during Skeletal Muscle Development in Rabbits

An average of 94.55 million clean reads were obtained from our libraries. The Q20 ratio, Q30 ratio, and GC content of the clean reads ranged from 98.00% to 98.47%, 94.59% to 95.66%, and 51.00% to 59.00%, respectively (Appendix A). The whole-transcriptome sequencing data provided mRNA expression profiles. A differential analysis showed that many mRNAs were significantly changed during muscle development. In detail, 1963 and 2061 mRNAs were upregulated and downregulated from D0 to D35, respectively (Appendix A), and 1743 and 1548 mRNAs were upregulated and downregulated from D35 to D70, respectively (Appendix A). Three key transcription factors (TFs) of skeletal muscle development, including *MYOG*, *MYF5*, and *MEF2C*, were upregulated from D0 to D35 and downregulated from D35 to D70 (Figure 1A).

K-means clustering sorted differentially expressed mRNAs (DEmRNAs) into five groups (mRC1-mRC5, Figure 1B). The mRNAs in mRC2 were lowly expressed in D35 and D70, while they were highly expressed in D0, representing the D0-selective mRNAs. A GO analysis of mRNAs in mRC2 showed that skeletal muscle tissue development, muscle structure development, and skeletal muscle organ development were the top three significantly enriched in gene ontology in biological process category (GO-BP) terms. The KEGG pathway analysis showed that the top three significantly enriched signaling pathways by the mRNAs in mRC2 were the MAPK signaling pathway, arrhythmogenic right ventricular cardiomyopathy, and hypertrophic cardiomyopathy (Figure 1C). The mRNAs in mRC1 and mRC3 were lowly expressed in D0 and D70, while they were highly expressed in D35, representing the D35-selective mRNAs. A GO analysis of mRNAs in mRC1 and mRC3 showed that nervous system development, cell division, and cell cycle process were the top three significantly enriched GO-BP terms. A KEGG pathway analysis showed that the top three significantly enriched signaling pathways by the mRNAs in mRC1 and mRC3 were axon guidance, ECM-receptor interaction, and cell adhesion molecules (Figure 1D). The mRNAs in mRC4 were lowly expressed in D0 and D35, while they were highly exclusively expressed in D70, representing the D70-selective mRNAs. GO enrichment showed that the top three significantly enriched GO-BP terms by the mRNAs in mRC4 were regulation of cytokine production, cytokine production, and immune response (Figure 1E). The KEGG pathway analysis showed that the top three significantly enriched signaling pathways by the mRNAs in mRC4 were leukocyte transendothelial migration, neuroactive ligand-receptor interaction, and osteoclast differentiation (Figure 1E). On the other hand, D35- and D70-selective mRNAs were also enriched in muscle development-related GO-BP terms (Appendix A).

### 3.2. Identification and Characterization of circRNAs in Rabbit Skeletal Muscles

By exploring the whole-transcriptome sequencing libraries, we found approximately 8% circular splicing events in the libraries, presenting a large opportunity for circRNA discovery (Figure 2A). By performing circRNA detection, we identified 28,407 circRNAs. To obtain well-assembly circRNA sequences, we performed circRNA-seq for the pooled sample used in whole-transcriptome sequencing and assembled 9253 circRNAs. Based on the back splicing junction (BSJ) information, we merged the circRNAs in the two types of sequencing libraries and obtained 6845 credible circRNAs that were simultaneously detected in both types of libraries, which were used for a downstream analysis (Figure 2B).

An analysis of genomic annotation found that most circRNAs (89.64%) were transcribed from the exons of protein-coding genes, and a small number of circRNAs were transcribed from the intron and intergenic regions (Figure 3A,C). Most circRNAs contained 2–5 exons (Figure 3B) and showed a 200–500 bp sequence length (Figure 3C). An analysis of circRNA expression found that 2898 circRNAs were expressed in all three stages (Figure 3D). Based on the sequences of circRNAs identified in our samples and circRNAs of mice and humans retrieved from public databases, we further explored circRNA conservation. Interestingly, our data showed a higher number of conserved circRNAs between rabbits and humans than between rabbits and mice. In detail, 1234 circRNAs were conserved between rabbits and mice, 3423 circRNAs were conserved between rabbits and humans, and 1150 circRNAs were conserved among the three species. On the other hand, there were 84 and 2273 rabbit circRNAs exclusively conserved with mice and humans, respectively (Figure 3E).

### 3.3. Differential Analysis of circRNAs during Skeletal Muscle Development in Rabbits

Hierarchical clustering based on the CPM values sorted nine samples into three distinct clusters corresponding to D0, D35, and D70, indicating the excellent reproducibility of our whole-transcriptome sequencing data (Figure 4A). To understand the regulation of circRNAs in skeletal muscle development, we performed a differential analysis. Our results showed that 176 and 420 circRNAs were significantly downregulated and upregulated from D0 to D35, respectively (Figure 4B). On the other hand, 281 and 138 circRNAs were significantly downregulated and upregulated from D35 to D70, respectively (Figure 4C). The Venn diagrams constructed using differentially expressed circRNAs (DECs) showed no overlap between the upregulated circRNAs from D0 to D35 and the upregulated circRNAs from D35 to D70, and no overlap between the downregulated circRNAs from D0 to D35 and the downregulated circRNAs from D35 to D70. On the other hand, 85 downregulated circRNAs from D0 to D35 were upregulated from D35 to D70, and 212 upregulated circRNAs from D0 to D35 were downregulated from D35 to D70 (Figure 4D).

### 3.4. The Expression Relationships between circRNAs and Their Hose Genes

The DECs and DEmRNAs were subjected to an expression relationship analysis between circRNAs and their host genes. From D0 to D35, 34 and 113 circRNAs were synchronously changed with their downregulated and upregulated host genes, respectively (Figure 5A). GO enrichment showed that neural tube development, secondary metabolic process, and retrograde transport and endosome to Golgi were the top three significantly enriched GO-BP terms by the host genes of those upregulated circRNAs. A KEGG pathway analysis showed that the MAPK signaling pathway, Ras signaling pathway, and Rap1 signaling pathway were the top three significantly enriched signaling pathways by the host genes of those upregulated circRNAs (Figure 5B). From D35 to D70, 57 and 29 circRNAs were synchronously changed with their downregulated and upregulated host genes, respectively (Figure 5A). GO enrichment showed that the polysaccharide biosynthetic process, regulation of signaling receptor activity, and polysaccharide metabolic process were the top three significantly enriched GO-BP terms by the host genes of those upregulated DECs. A KEGG pathway analysis showed that the choline metabolism in cancer, fatty acid biosynthesis, and selenocompound metabolism were the top three significantly enriched signaling pathways by the host genes of those upregulated circRNAs (Figure 5C). In contrast to those synchronously changed between DECs and DEmRNAs, there were eight circRNAs that were oppositely changed with their host genes from D0 to D35, including four upregulated circRNAs transcribed from *ZBTB40*, *AGL*, *FBXW4*, and *PCMT1* and four downregulated circRNAs transcribed from *ATP11B*, *SRBD1*, *ENSOCUG00000017307*, and *TTC3*. There were three circRNAs that were oppositely changed with their host genes from D35 to D70, including one upregulated circRNA transcribed from *PGM2* and two downregulated circRNAs transcribed from *AGL* and *COQ9* (Figure 5A).

### 3.5. Construction of the DECs Involved Putative ceRNA Network

To better understand the RNA regulatory networks during skeletal muscle development in rabbits, we retrieved differentially expressed miRNAs during skeletal muscle development in rabbits in one previous study [30] and identified putative DEC-DEmiRNA interactions using miRanda software. We obtained 1594 DEC–DEmiRNA interaction pairs from D0 to D35 (Figure 6A and Appendix A). An analysis of DEC–DEmiRNA interaction pairs found a positive relationship between circRNA lengths and the number of their interacted miRNAs (Figure 6B). The top five circRNAs that had the highest number of interacted miRNAs were novel_circ_0017312, novel_circ_0012335, novel_circ_0013404, novel_circ_0014377, and novel_circ_0016614 (Figure 6C).

By predicting the differentially expressed target mRNAs of the miRNAs in the DEC–DEmiRNA interaction pairs, we obtained putative DEC–DEmiRNA–DEmRNA networks, in which the circRNAs and mRNAs had synchronous expression changes from D0 to D35 (Appendix A). Notably, nine circRNAs were predicted to regulate their host genes by sponging miRNAs, including the circRNAs transcribed from *PTPRD*, *NECTIN3*, *MIGA1*, *HDGFL3*, *TMTC2*, *BOAT2*, *FGFR4*, *POLQ*, and *CLCN1* (Figure 7A). An analysis of the DEC–DEmiRNA–DEmRNA networks found that the upregulated circRNAs from D0 to D35 primarily functioned as regulators of the genes involved in neuron development-related GO-BP terms, such as neuron development, nervous system development, and neuron differentiation (Figure 7B).

On the other hand, we obtained 1157 DEC–DEmiRNA interaction pairs from D35 to D70 (Figure 8A and Appendix A). An analysis of DEC–DEmiRNA interaction pairs found a positive relationship between circRNA lengths and the number of their interacted miRNAs (Figure 8B). The top five circRNAs that had the highest number of interacted miRNAs were novel_circ_0017312, novel_circ_0012335, novel_circ_0013404, novel_circ_0002967, and novel_circ_0016614 (Figure 8C).

By predicting the differentially expressed target mRNAs of the miRNAs in the DEC–DEmiRNA interaction pairs, we obtained putative DEC–DEmiRNA–DEmRNA networks, in which the circRNAs and mRNAs had synchronous expression changes from D35 to D70 (Appendix A). Five circRNAs were predicted to regulate their host genes by sponging miRNAs, including the circRNAs transcribed from *CLCN1*, *PTPRD*, *TASP1*, *FGFR4*, and *POLQ* (Figure 9A). An analysis of the DEC–DEmiRNA–DEmRNA networks found that the upregulated circRNAs from D35 to D70 primarily functioned as regulators of the genes involved in metabolic processes-related GO-BP terms, such as the glycogen metabolic process, cellular glucan metabolic process, and glucan metabolic process (Figure 9B). Furthermore, the upregulated circRNAs from D35 to D70 in the DEC–DEmiRNA–DEmRNA networks were also significantly enriched in metabolic processes-related KEGG pathways, such as insulin signaling and glucagon signaling pathways (Figure 9B).

## 4. Discussion

In this study, our data revealed the mRNA and circRNA expression changes from newborn to weaning and weaning to puberty. Weaning is a crucial point in rabbit production, during which the sources of nutrition absorption of rabbits change from milk to an ordinary diet. The MRF and MEF2 families have been intensively reported to regulate the development of skeletal muscle [35,36]. The mRNA analysis showed that MRF members *MYOG* and *MYF5* and MEF2 member *MEF2C* were significantly changed during the development of skeletal muscle and were upregulated from newborn to weaning in rabbits, suggesting that the period from newborn to weaning is critical for forming TF-mediated gene programming during skeletal muscle development. The skeletal muscle is a highly heterogeneous tissue containing many different and dynamic cell types, such as myocyte, satellite cells, fibro-adipogenic progenitors, and immune cells [37]. Using the K-means clustering method, we obtained stage-selective mRNAs at the newborn, weaning, and puberty stage. Our data showed that many skeletal muscle development-related biological processes were significantly enriched by the D0-, D35-, and D70-selective mRNAs. On the other hand, stage-selective mRNAs were also responsible for the different biological processes during skeletal muscle development in rabbits. For instance, the D0-selective mRNAs were mainly enriched in muscle development-related biological processes, the D35-selective mRNAs mainly functioned in nervous development-related biological processes, and the D70-selective mRNAs were enriched in the biological processes associated with cytokine production and immune response, suggesting the sequential gene programming for functional diversity improvement during skeletal muscle development in rabbits.

Based on the BSJ reads, previous studies have identified many circRNAs using whole-transcriptome data [37,38], while the existence of both linear transcripts and circular transcripts was reported to increase the degree of difficulty in distinguishing sequencing reads from linear RNA and circular RNAs when performing the transcript assembly of circRNAs [27]. The removal of linear RNA during the construction of circRNA-seq libraries can improve the accuracy of circRNA identification and assembly. To make the most of our whole-transcriptome data and reduce the cost of circRNA-seq, we assembled circRNAs using circRNA-seq of pooled samples subjected to whole-transcriptome sequencing. Then, we merged the circRNAs identified in these two types of libraries to reduce the false positive rate of circRNAs. The characterizations of most circRNAs being transcribed from exons, being short length sizes, and with fewer exon numbers were similar to those identified in other mammals, such as pigs [39], cattle [40], and goats [41], indicating our circRNA data were credible. Previous studies have reported that circRNAs are a class of evolutionarily conserved RNA molecules [7]. In this study, we compared the sequences of rabbit circRNAs to the mouse circRNAs and human circRNAs. We found that many rabbit circRNAs are conserved among humans, mice, and rabbits, which was in line with previously identified circRNAs in pigs [13] and sheep [42]. On the other hand, there are many more conserved circRNAs between rabbits and humans than between rabbits and mice, which might indicate the feasibility of rabbits acting as human muscle models in studying circRNA functions. A differential analysis of circRNA showed no circRNA continuously upregulated or downregulated from newborn to puberty, suggesting that the expression of stage-specific circRNA regulates skeletal muscle development in rabbits.

Currently, our knowledge of the regulatory relationship between circRNAs and host genes remains limited. Recent studies revealed that circRNAs could terminate and promote the mRNA transcription of the host gene by R-loop and ceRNA mechanisms, respectively [11,12]. Considering the limited data in this study, we compared the expression changes between circRNAs and host genes to predict the potential regulation of circRNAs in host genes. Most circRNAs were synchronously changed with their host genes, which might indicate circRNAs promoting the expression of host genes. The upregulated mRNAs of host genes with upregulated circRNAs were predicted to be enriched in key muscle development-related signaling pathways, such as MAPK signaling pathways [43] from D0 to D35, and fatty acid biosynthesis [44] from D35 to D70, suggesting the potential functions of circRNAs by regulating host genes during skeletal muscle development in rabbits. Interestingly, 11 circRNAs that were oppositely changed with their host genes were identified, which potentially indicated that circRNAs inhibited their host genes. However, whether these circRNAs were inhibiting the host genes by R-loop or other mechanisms warrants further investigation.

The circRNAs playing roles by acting as miRNA sponges were widely reported in previous studies [45]. In this study, we constructed circRNA–miRNA interaction networks. Our data showed that the circRNA with a longer length could interact with more different miRNAs, which suggests that long circRNAs might play more critical or complex roles during skeletal muscle development in rabbits than short circRNAs. On the other hand, an analysis of the target genes of miRNAs in circRNA–miRNA networks found several circRNAs could regulate the expression of host genes by sponging shared-interaction miRNAs. The *FGFR4* signaling pathway is necessary for limb muscle differentiation [46]. *FGFR4* was upregulated from newborn to weaning and was synchronously changed with its circRNA in this study. Our results showed that the circRNA transcribed from FGFR4 was predicted to regulate FGFR4 by sponging miR-185-3p, indicating that circRNAs might protect some essential muscle genes from degradation to promote skeletal muscle development. Nerve fiber development is important for skeletal muscle development [47,48]. From newborn to weaning, the improvement of the ability of limb movement was an obvious characteristic of the rabbits. Our data showed that the circRNAs primarily functioned as regulators of the genes involved in neuron development via DEC-DEmiRNA–DEmRNA networks from D0 to D35, indicating circRNAs were critical for the formation of motor function of skeletal muscle during the early development of rabbits. Skeletal muscle has recently been identified as a metabolic organ and many cytokines and peptides are produced, expressed, and released by muscle fibers [49,50], and studies have emphasized that skeletal muscle is essential for metabolism due to its role in insulin resistance and glucose uptake [50]. Our data showed that the circRNAs were regulators of genes that were significantly enriched in metabolic processes-related GO-BP terms, which might suggest that circRNAs regulated the formation of the metabolic functions of skeletal muscle from weaning to puberty in rabbits.

## 5. Conclusions

In conclusion, from our data it is possible to identify genome-wide circRNAs and their dynamics from newborn to weaning and puberty in rabbits. CircRNAs might regulate skeletal muscle development by regulating their host genes or miRNA-mediated ceRNA networks according to the corresponding growth stages. Our work provides a catalog of circRNAs regulating skeletal muscle development at key growth stages in rabbits and might give new insight into meat rabbit breeding.

## Figures and Tables

**Figure 1 animals-12-02208-f001:**
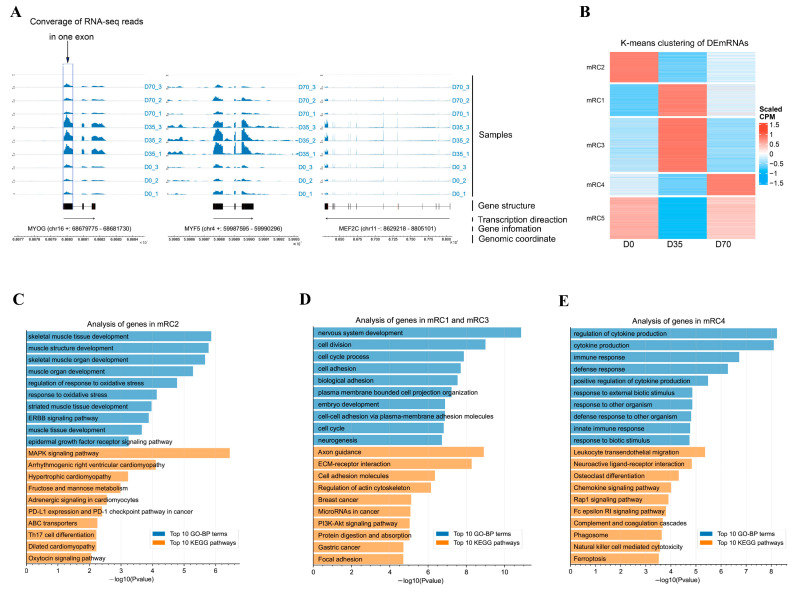
Dynamics of mRNA expression during development of skeletal muscle tissues at three growth stages in rabbits. (**A**) The RNA-seq tracks of all samples for the key genes (*MYOG*, *MYF5*, and *MEF2C*) of skeletal muscle development. Each track shows one sample. The blue signals represent the coverage of RNA-seq reads. The genomic regions are shown, and the selected genes are highlighted in black. The gene information was marked by ‘gene name (chromosome strand: start–end)’. The black arrows indicate the transcription directions. (**B**) K-means clustering analysis of differentially expressed mRNAs. The mRC1–mRC5 show the five sorted groups by K-means clustering. (**C**) GO enrichment and KEGG pathway analysis of mRNAs in mRC2. The top 10 significantly enriched GO-BP terms and top 10 significantly enriched KEGG pathways were shown. (**D**) GO enrichment and KEGG pathway analysis of mRNAs in mRC1 and mRC3. The top 10 significantly enriched GO-BP terms and top 10 significantly enriched KEGG pathways were shown. (**E**) GO enrichment and KEGG pathway analysis of mRNAs in mRC4. The top 10 significantly enriched GO-BP terms and top 10 significantly enriched KEGG pathways were shown.

**Figure 2 animals-12-02208-f002:**
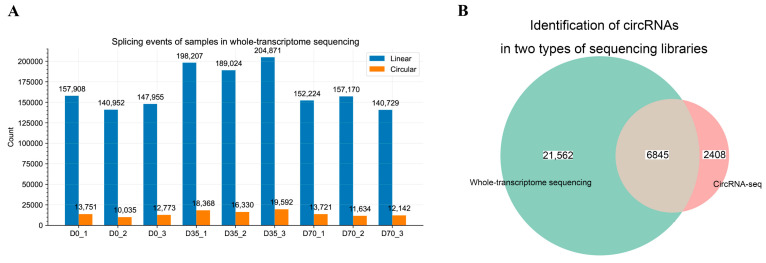
Identification of circRNAs in skeletal muscle tissues at three growth stages of rabbits. (**A**) Splicing events were detected in each skeletal muscle tissue by whole-transcriptome sequencing. (**B**) The Venn diagram of circRNAs from whole-transcriptome sequencing libraries and circRNA-seq library.

**Figure 3 animals-12-02208-f003:**
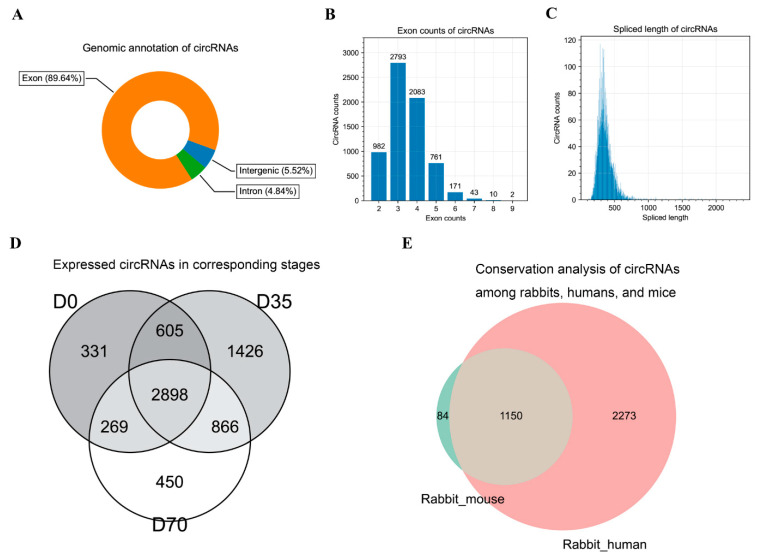
Characterization of circRNAs in skeletal muscle tissues. (**A**) Genomic distribution of circRNAs. (**B**) The number of circRNAs with different exon counts. (**C**) The length distribution of circRNAs. (**D**) Venn diagram of the expressed circRNAs in D0, D35, and D70. (**E**) The number of conserved circRNAs between rabbits and mice, and between rabbits and humans.

**Figure 4 animals-12-02208-f004:**
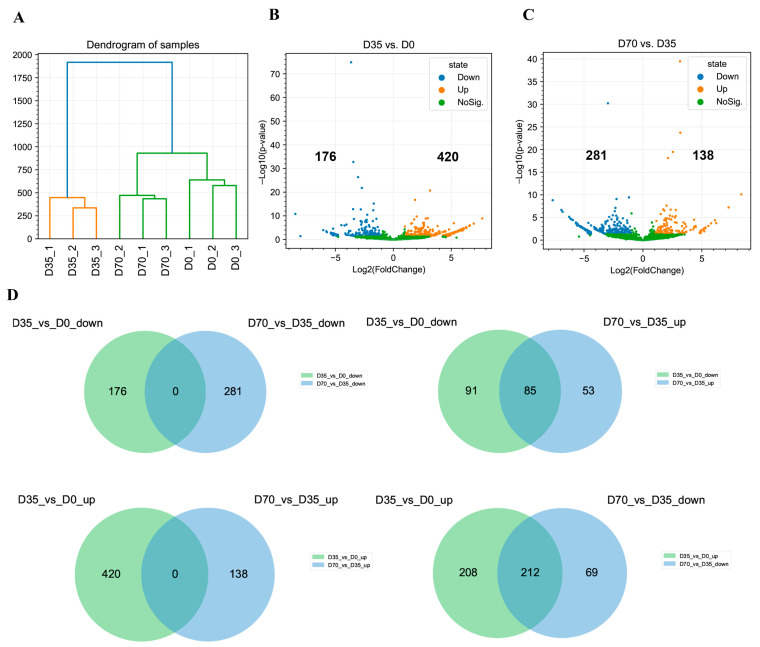
Differential analysis of circRNAs during skeletal muscle development of rabbits. (**A**) Hierarchical clustering analysis of nine samples with all identified circRNAs based on the CPM values. (**B**) Volcano plot of DECs in D35 vs. D0. (**C**) Volcano plot of DECs in D70 vs. D35. (**D**) Venn diagram showing the DECs at the two growth periods.

**Figure 5 animals-12-02208-f005:**
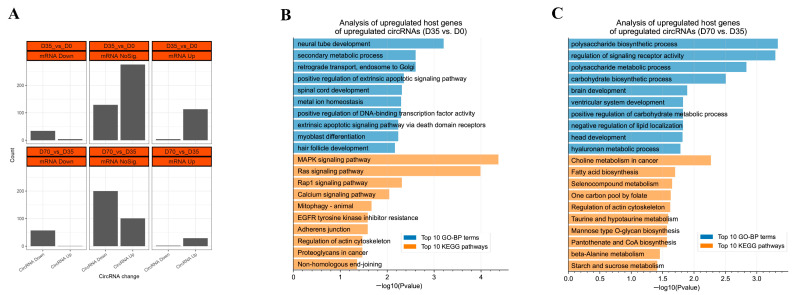
Expression relationships between circRNAs and their hose genes and functional annotation of upregulated circRNAs transcribed from upregulated host genes. (**A**) Classification of DECs based on the expression relationships between circRNAs and hose genes. (**B**) GO enrichment and KEGG analysis of upregulated DEmRNAs with upregulated DECs from D0 to D35. (**C**) GO enrichment and KEGG analysis of upregulated DEmRNAs with upregulated DECs from D35 to D70.

**Figure 6 animals-12-02208-f006:**
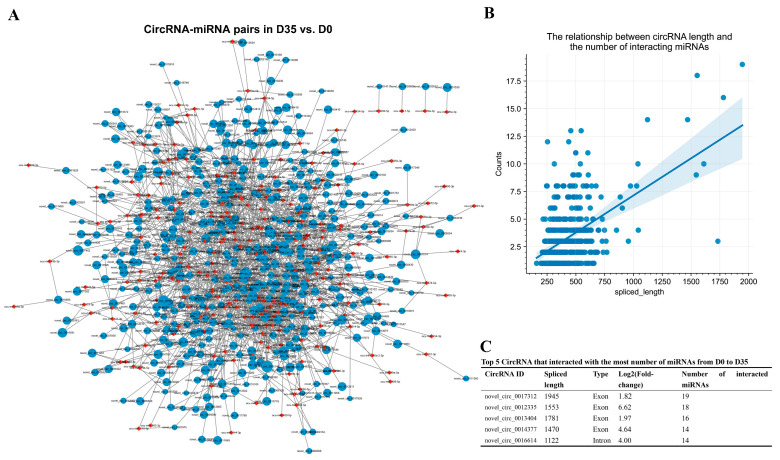
Prediction of putative circRNA–miRNA interaction pairs from D0 to D35. (**A**) Prediction of putative circRNA–miRNA interaction pairs from D0 to D35. The blue and red nodes indicate the circRNAs and miRNAs, respectively. The sizes of blue nodes indicate the values of |log2(fold-change)|. (**B**) The relationship between circRNA lengths and numbers of interacted miRNAs from D0 to D35. (**C**) The top 5 circRNAs that interact with the highest numbers of miRNAs were shown.

**Figure 7 animals-12-02208-f007:**
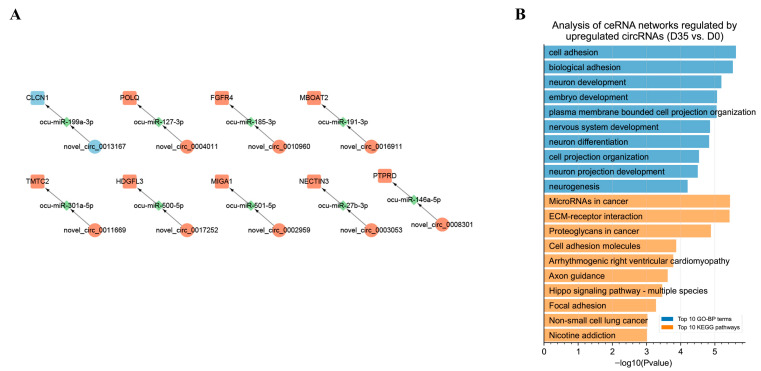
Functional annotation of circRNAs in putative DEC–DEmiRNA–DEmRNA networks from D0 to D35. (**A**) From D0 to D35, the circRNAs regulate host genes by interacting with miRNAs. The red and blue nodes indicate the upregulated and downregulated genes, respectively. (**B**) GO enrichment and KEGG pathway analysis of upregulated mRNAs in DEC–DEmiRNA–DEmRNA network from D0 to D35.

**Figure 8 animals-12-02208-f008:**
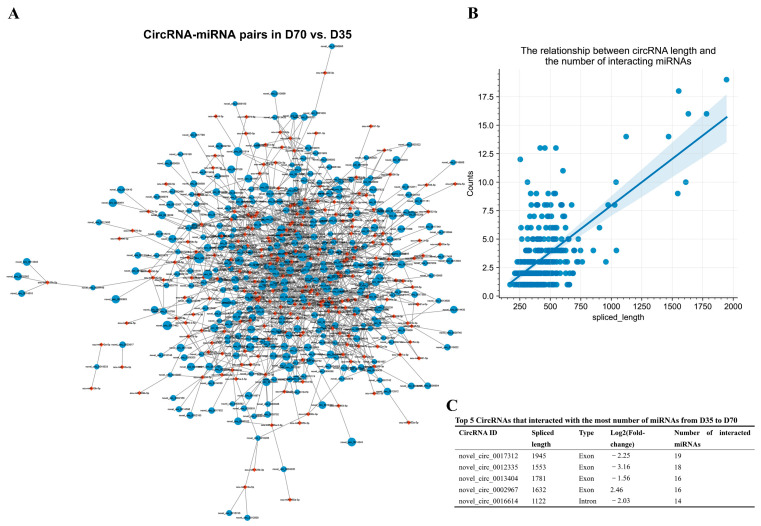
Prediction of putative circRNA–miRNA interaction pairs from D35 to D70. (**A**) Prediction of putative circRNA–miRNA interaction pairs from D35 to D70. The blue and red nodes indicate the circRNAs and miRNAs, respectively. The sizes of blue nodes indicate the values of |log2(fold-change)| in D70 vs. D35. (**B**) The relationship between circRNA lengths and numbers of interacted miRNAs from D35 to D70. (**C**) The top 5 circRNAs that interact with the highest numbers of miRNAs were shown.

**Figure 9 animals-12-02208-f009:**
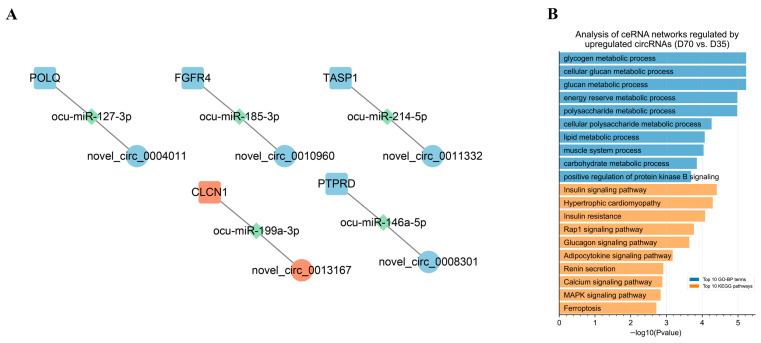
Functional annotation of circRNAs in putative DEC–DEmiRNA–DEmRNA networks from D35 to D70. (**A**) From D35 to D70, the circRNAs regulate host genes by interacting with miRNAs. The red and blue nodes indicate the upregulated and downregulated genes, respectively. (**B**) GO enrichment and KEGG pathway analysis of upregulated mRNAs in DEC–DEmiRNA–DEmRNA network from D35 to D70.

## Data Availability

Not applicable.

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
