# Peer review of "Genome-Wide Identification and Characterization of Circular RNAs during Skeletal Muscle Development in Meat Rabbits"

_animals, 2022, doi:10.3390/ani12172208_

Round 1

Reviewer 1 Report

There are repeated words in the title and keywords. Authors must choose where to leave them and remove the repeated ones.

Article 1

Genome-wide identification and characterization of circular RNAs during skeletal muscle development in meat rabbits

Keywords: Muscles; meat rabbits; RNA-seq; circRNAs; meat

In lines 174 at 176:- It is Material and method, that need to change position.:-

 [In this study, nine whole-transcriptome sequencing libraries from skeletal muscle tissues of 0 (D0), 35 (D35), and 70 days-old (D70) ZIKA rabbits were constructed (n = 3 per group), which represented newborn, weaning, and puberty stage, respectively.]

Supplementary Materials:- Authors must include a link that takes the reader to Supplementary Materials: As it is, it is not indicated where to find the material.

Not possible to see the table (in Supplementary Materials). I didn’t receive it.

In line 205 is correct two times?: … cytokine production, cytokine production, and immune response.????

The figure 1, no one possibly understands the images. Need to adjust. Each image must be clear and self-explanatory. It is not possible to understand each of the figures. Bundled up like this they contribute little or nothing! All piled up.

Figure 2. Figure two has interesting information, but it's all crammed together and small. Poorly readable - poor visualization. As the information is described in lines 219 to 238, you should choose how to show it. I prefer the figures and then discuss them. But as they are, they repeat.

The same is true of figure 3. A bunch of images crammed together, with bad visualization.

Again, in Figure 5, it is not possible to interpret what the authors intend to show. Small and confusing figures.

5. Conclusions

 In line 435: In conclusion, our data Is possible to identify genome-wide circRNAs and their dynamics from……..

Author Response

Reviewer #1

  1. There are repeated words in the title and keywords. Authors must choose where to leave them and remove the repeated ones

Article 1                                          

Genome-wide identification and characterization of circular RNAs during skeletal muscle development in meat rabbits

Keywords: Muscles; meat rabbits; RNA-seq; circRNAs; meat

>>> We thank the reviewer for this important note. Based on the reviewer’s comment, we removed repeat words in Keywords and added new unique keywords. Please see our revised manuscript (line 38 and 39).

  1. In lines 174 at 176:- It is Material and method, that need to change position.:- [In this study, nine whole-transcriptome sequencing libraries from skeletal muscle tissues of 0 (D0), 35 (D35), and 70 days-old (D70) ZIKA rabbits were constructed (n = 3 per group), which represented newborn, weaning, and puberty stage, respectively.]

>>> We thank the reviewer for the good suggestion. We removed the sentence in our revised manuscript and added it on the section of Material and method. Please see our revised manuscript for details (line 86 and 92).

  1. Supplementary Materials:- Authors must include a link that takes the reader to Supplementary Materials: As it is, it is not indicated where to find the material. Not possible to see the table (in Supplementary Materials). I didn’t receive it.

>>> We thank the reviewer for this note. We had uploaded all related “Supplementary Materials” in our original manuscript in ZIP file format. May be unknown reasons of Animals system, you missed the file. Please contact journal editor to review our “Supplementary Materials”.

  1. In line 205 is correct two times?: … cytokine production, cytokine production, and immune response.????

>>> We thank the reviewer for this comment and regret the confusion between the two GO-BP terms of “regulation of cytokine production” and “cytokine production”. Actually, they are different GO terms in functional annotation (also can be found in Figure 1E). To clearly show this result, we added “Figure 1E” in the end of this sentence to guide reader to understand. Please see our revised manuscript for details (line 207 and 208).

  1. The figure 1, no one possibly understands the images. Need to adjust. Each image must be clear and self-explanatory. It is not possible to understand each of the figures. Bundled up like this they contribute little or nothing! All piled up.

>>> We thank the reviewer for this suggestion. To better understand the figures for readers, we carefully annotated Figure 1A (Because Figure 1A seems more complex among subfigures in Figure 1). On the other hand, we added a title for Figure 1B. Finally, we gave a more detailed figure legend for each subfigure in our revised manuscript. Please see our revised manuscript for details (Figure 1 and line 216 - line 228).

  1. Figure 2. Figure two has interesting information, but it's all crammed together and small. Poorly readable - poor visualization. As the information is described in lines 219 to 238, you should choose how to show it. I prefer the figures and then discuss them. But as they are, they repeat.

The same is true of figure 3. A bunch of images crammed together, with bad visualization.

Again, in Figure 5, it is not possible to interpret what the authors intend to show. Small and confusing figures.

>>> We thank the reviewer for this important note. As per the reviewer’s suggestion, we zoomed and restructured the subfigures in Figure 2, 3, and 5 in our revised manuscript, respectively. Also gave a more detailed figure legend for these figures. Please see our revised manuscript for details (Figure 2, 3, 4, 6, 7, 8 and 9).

  1. In line 435: In conclusion, our dataIs possible to identify genome-wide circRNAs and their dynamics from……..

>>> We thank the reviewer for this important note in “Conclusion” section. Based on the reviewer’s suggestion, we revised this sentence as “In conclusion, our data is possible to identify genome-wide circRNAs and their dynamics from …” in our revised manuscript to make a robust conclusion. Please see our revised manuscript for details (line 484).

Reviewer 2 Report

Dear Authors,

I would like to applaud your effort with your study, which are very intresting and with high novelity.

Hovewer there are few concerns within manuscript:

-lines 63-67 - please add some citation to support what you qrote

-lines 67 - 74 I don't think this part is sufficient to introduction. This is summary and I think this part should be removed.

line 84 - which study, please provide citation, ad libitum should be italic. Also day 0 (D0) is just after birth?

lines 85 - to be honest this part raised my concerns - I never heard of this method - injection of air vein- and after search i found that this method is unacceptable in livestock. Can authors comment this and provide explanation/publications which support choose of this method please.

 Figure 5 A and E - maybe this two pictures move to separate sheet - it is so small that only blue and red dots blurred are visible.

Author Response

Reviewer #2

  1. -lines 63-67 - please add some citation to support what you qrote

>>> We thank the reviewer for this important note. Based on the reviewer’s comments, we added 3 references for the two sentences. Please see our revised manuscript for details (line 64 and 65).

  1. -lines 67 - 74 I don't think this part is sufficient to introduction. This is summary and I think this part should be removed.

>>> We thank the reviewer for this important suggestion. We removed relative sentences in introduction section. Please see our revised manuscript for details.

  1. line 84 - which study, please provide citation, ad libitumshould be italic. Also day 0 (D0) is just after birth?

>>> We thank the reviewer for these important suggestions and regret for this citation missing. We added reference for the description of standard diet. Please see line 85 in our revised manuscript. The “ad libitum” were italic in our revised manuscript. Yes, D0 represents newborn rabbits that were just after birth. We gave this description in line 85 in our revised manuscript.

  1. lines 85 - to be honest this part raised my concerns - I never heard of this method - injection of air vein- and after search i found that this method is unacceptable in livestock. Can authors comment this and provide explanation/publications which support choose of this method please.

>>> We thank the reviewer for this important note. The rabbit is special in livestock species because it often acts as an experimental animal, e.g. medical model animals. The sacrifice method of rabbits might be diverse in different studies. The injection of air vein was used in many rabbit studies, such as studies conducted by Zheng (New Zealand rabbits, DIO: doi: 10.1186/s12864-021-07896-5), Li (Japanese white rabbits, DIO: 10.3760/cma.j.issn.2095-4352.2015.04.005), and Chen (Japanese white rabbits, 10.13618/j.issn.1001-5728.2008.03.010). Therefore, our study adopts the method of injecting air into the ear vein of rabbits.

  1. Figure 5 A and E - maybe this two pictures move to separate sheet - it is so small that only blue and red dots blurred are visible.

>>> We thank the reviewer for this important suggestion. As per the reviewer’s suggestion, we move Figure 5 A and E to separate sheet and zoomed them. On the other hand, considered the complexity of networks in our or other studies, the large networks often loss information in figures (duo to the limited size of page). Therefore, we provided the original data of networks in table format in Supplementary Materials of revised manuscript. Thus, readers can access the detailed information for these networks. Please see the revised manuscript for details (Figure 6 - 9, Table S5, and Table S7).

Round 2

Reviewer 2 Report

Dear Authors,

Thank you very much for your response and congratulation on your research